# Mycetoma epidemiology, diagnosis management, and outcome in three hospital centres in Senegal from 2008 to 2018

**Doudou Sow**[1,2,3]*, **Maodo Ndiaye**[4], **Lamine Sarr**[5], **Mamadou D. Kanté**[4], **Fatoumata Ly**[6], **Pauline Dioussé**[7], **Babacar T. Faye**[2], **Abdou Magip Gaye**[8], **Cheikh Sokhna**[3], **Stéphane Ranque**[9], **Babacar Faye**[2]

**1** Service de Parasitologie-Mycologie, UFR Sciences de la Santé, Université Gasbon Berger de Saint Louis, Saint Louis, Sénégal, **2** Service de Parasitologie-Mycologie, Faculté de médecine, Université Cheikh Anta Diop de Dakar, Dakar, Sénégal, **3** UMR VITROME, Campus International IRD-UCAD de l'IRD, Dakar, Sénégal, **4** Service de Dermatologie, Hôpital Aristide Le Dantec, Dakar, Sénégal, **5** Service d'Orthopédie, Hôpital Aristide Le Dantec, Dakar, Sénégal, **6** Service de Dermatologie, Institut d'Hygiène Sociale, Dakar, Sénégal, **7** Service de Dermatologie, Centre Hospitalier Régional de Thiès, Thiès, Sénégal, **8** Service d'anatomie pathologie, Hôpital Aristide Le Dantec, Dakar, Sénégal, **9** Aix Marseille Univ, IRD, APHM, SSA, VITROME, IHU-Méditerranée Infection, Marseille, France

* doudou.sow@ugb.edu.sn

**Data Availability Statement:** All relevant data are within the paper.

**Funding:** The authors received no specific funding for this work.

## Abstract

Mycetoma is a neglected tropical disease caused by various actinomycetes or fungi. The disease is characterized by the formation of tumor like-swellings and grains. Senegal is an endemic country where mycetoma cases are under-or misdiagnosed due to the lack of capacities and knowledge among health workers and the community; and where the management of eumycetoma, burdened by a high amputation rate, is currently inadequate. This study aimed to update data on the epidemiology of mycetoma cases diagnosed in three hospital centres in Senegal over a 10 years-period. A total of 193 patients, diagnosed from 2008 to 2018, were included in the study. The most frequent presentation was eumycetoma (47.2%); followed by actinomycetoma (36.8%); it remained undetermined in 16.1% of the patients. The mean age was 38.3 years (68.4% of the patients were between 15 and 45 years-old); the male: female ratio was a 2.94; and most were farmers. One hundred fifty-six (80.8%) patients had used phytotherapy before attending the hospital. Mycetoma was mainly located to the lower limbs (91.2%). Grains were observed in 85% of the patients; including white (25.6%) and yellow (4.3%) grains. The etiological diagnosis was complex, resulting in negative direct microscopy, culture and/or histopathology findings, which explains that 16.1% remained uncharacterized. In most of cases, actinomycetoma were treated with a combination of cotrimoxazole, amoxicillin/clavulanic acid, and streptomycin; whereas eumycetoma cases were treated with terbinafine. The surgery was done in 100 (51.8%) of the patients including 9 in actinomycetoma, 78 in eumycetoma and 13 in undetermined form. The high number of uncharacterized mycetoma in this study, the delay in attending a qualified health-care facility, and the lack of available adequate antifungal drug, point out the need to strengthen mycetoma management capacities in Senegal.

**Competing interests:** The authors have declared that no competing interests exist.

## Introduction

Mycetoma is a tropical chronic granulomatous disease, with the formation of tumor-like soft tissue swelling and the formation of grains [1]. This disease usually results from small traumatic implantation of causative agent in subcutaneous tissue mainly located to the foot [2]. The disease is responsible of massive distortion, deformities and disabilities, and can be life-threatening if not adequately managed [3]. The involved infectious agents are either aerobic filamentous actinomycetes, causing actinomycetoma, or filamentous fungi, causing eumycetoma. Actinomycetoma are commonly caused by *Streptomyces somaliensis* and *Nocardia* spp. while *Madurella mycetomatis* is the commonest agent in eumycetoma [4–6].

The treatment of mycetoma cases depends on whether the causative agent is an actinomycete or a fungus. Whereas actinomycetoma are usually adequately treated with antibacterials, eumycetoma show usually a relatively poor response to available antifungal drugs. Therefore, surgery is recommended as the best treatment option of eumycetoma. The surgical treatments ranges from broad surgical excision to limb amputation [7,8].

Mycetoma was recently added to the WHO list of Neglected Tropical Diseases (NTD) following the Geneva meeting in May 2016. This recognition has brought much more attention to this disease. However, the global burden of mycetoma is still unknown due to the absence of case reporting system [1,9]. A recent meta-analysis has estimated a mycetoma prevalence up to 1.81 and 3.49 cases per 100,000 habitants in Sudan and Mauritania, respectively [2]. Senegal, a country located within the « mycetoma belt » area is burdened with a relatively high mycetoma prevalence. Some studies have described the profile of actinomycetoma and eumycetoma cases diagnosed in the country through cases series a decade ago [10–12]. The most recent papers have mainly focused on case reports and have described the epidemiology in a single heath care centre [13,14]. This study aims to provide updated data on the epidemiology, clinical presentation, laboratory diagnosis, and treatment of mycetoma, based on the patients who had consulted in three health care facilities located in two regions of Senegal.

## Materials and methods

This descriptive, retrospective case series study was carried out between January 2008 and December 2018, in three hospital centres, including Le Dantec university hospital and the Institut d'Hygiene Sociale hospital, which are both located in the capital city Dakar, and the regional hospital of Thies located 70 km East from Dakar. The clinical records of 193 patients with mycetoma who visited the Dermatology wards of these hospitals and/or the orthopaedic ward of Le Dantec hospital, during the study period were thoroughly reviewed.

Mycetoma diagnosis criteria were compatible clinical examination findings (tumor like-swellings, presence of sinus tracts, discharge of pus and/or grains) and, when available, a laboratory diagnosis confirmation.

Clinical diagnosis was mainly based on the colour of the grain and the evolution of the lesion. The microbiological aspects included: 1) the direct microscopic examination of the specimens or the grains, in saline or cotton blue solution after adding 30% potassium hydroxide (KOH), and 2) the culture on both Sabouraud dextrose agar plus chloramphenicol and Lowenstein- Jensen agar. Biopsies have been made in each case for histological examination after Periodic acid–Schiff (PAS), Grocott's methenamine silver (GMS) and hematoxylin and eosin (H&E) staining. X-rays, ultrasound, and CT scan examinations have been performed to investigate severe lesions with bone involvement. At the time of this study, molecular diagnosis was not available at any of the centres. Therefore, species identification could not be confirmed molecularly.

Medical treatment consisted of trimethoprim-sulfamethoxazole, or a combination of tri-methoprim-sulfamethoxazole, streptomycin, and amoxicillin/clavulanic acid for actinomyce-toma and terbinafine or itraconazole for eumycetoma. In case it was not known if the lesion was an actino- or an eumycetoma, patients were treated with a combination of trimethoprim-sulfamethoxazole, streptomycin, amoxicillin/clavulanic acid and/or itraconazole.

Eumycetoma were also treated surgically. The surgical treatment consisted either in a cura-tive surgical resection of the lesion or a limb amputation in case of bone involvement. Lymph-adenopathy surgery consisted of lymph node dissection followed by histopathological examination.

The patients' demographic, place of residency, clinical and biological data were entered into excel table and statistical analysis was performed using TM R2.15.0 software (R Founda-tion for Statistical Computing, Vienna, Austria). Categorical variables were described as per-centages while mean and standard error were used for continuous variables.

### Ethical statement

This study has been approved by the Ethics Committee of the Cheikh Anta Diop University (Ref number: 0237/2017/CER/UCAD). All the data collected from the study participants have been de-identified. Informed consent was not required as the study consisted in retrospective review of the files of patients who had been diagnosed and treated for mycetoma.

## Results

A total of 193 patients diagnosed with mycetoma have been included in this study. The major-ity of these patients were recruited in Dakar, 108 (55.9%) at Le Dantec hospital, 53 (27.5%) at the Institut d'Hygiène Sociale hospital, and 32 (16.6%) at the Thies regional hospital. The mean age of the patients was 38.3±16.4 years (range: 8–84 years). Most of the patients were 30 years old or more as presented in Table 1; 74.6% of them were male, with a 2.94 male/female ratio (Table 1). Regarding their occupation, 120 (62.1%) were farmers; 22 (11.4%) housewives; 21(10.9%) shopkeeper; and 30 (15.5%) had another occupation, including 7 students and 10 unemployed persons.

The geographic distribution across Senegal of the mycetoma cases herein analysed is illus-trated in Table 1. Most of the patients originated from the central and northern regions of the country including 47 cases from Thies (24.4%), 31 from Diourbel (16.1%), 23 from Louga (11.9%) and 20 from St Louis (10.4%). Patients originating from the capital city Dakar accounted for 34 cases (17.6%); the remaining patients were distributed in the East and South-ern part of Senegal (Fig 1). The mycetoma cases were classified into three groups according to the type of grains and laboratory criteria: there were 71 (36.8%) actinomycetoma, 91 (47.2%) eumycetoma, and 31 (16.1%) undetermined form (Table 1). Noteworthy, there were more eumycetoma cases in the northern regions than in the other part of the country (Fig 1).

At presentation, the duration of the lesions varied according to cases. The majority of patients 90 (46.6%) had mycetoma for 1 to 5 years while 37 (19.1%) had the infection for more than 10 years (Table 2). In 102 cases (52.8%) the patients experienced pain. In the majority of these patients a painkiller was added to their treatment. The clinical form, the size of the lesion, the type of grains, the anatomical localization (Figs 2 and 3), the lymphatic dissemination and the bone lesion are presented in Table 2.

Diagnosis of mycetoma was confirmed in 97 cases (50.3%) using microbiological examina-tions and/or histology. The culture has isolated only 56 species (29%). *Actinomadura pelletieri* was the most isolated species (24) followed by *Madurella mycetomatis* (21), *Falciformispora senegalensis* (5), *Streptomyces somaliensis* (2), *Scedosporum apiospermum* (1), *Penicillium* sp.

Table 1. Demographic data of the study population.

| | Frequency | % |
|---|---|---|
| **Age distribution** | | |
| < 15 years | 4 | 2.1 |
| 15–30 Years | 59 | 30.6 |
| 30–45 Years | 62 | 32.1 |
| > 45 Years | 68 | 35.2 |
| **Gender** | | |
| Male | 144 | 74.6 |
| Female | 49 | 24.4 |
| Male: Female ratio | | 2.94 |
| **Occupation** | | |
| Farmer | 120 | 62.1 |
| Housewives | 22 | 11.4 |
| Street vendor | 21 | 10.9 |
| Other | 30 | 15.5 |
| **Locality of origin** | | |
| Thiès | 47 | 24.4 |
| Dakar | 34 | 17.6 |
| Diourbel | 31 | 16.1 |
| Louga | 23 | 11.9 |
| Saint-Louis | 20 | 10.4 |
| Matam | 10 | 5.2 |
| Kaolack | 6 | 3.1 |
| Fatick | 4 | 2.1 |
| Kaffrine | 3 | 1.6 |
| Sédhiou | 2 | 1 |
| Tamba | 2 | 1 |
| Kédougou | 1 | 0.5 |
| Kolda | 1 | 0.5 |
| Nouatchok | 1 | 0.5 |
| **Type of Mycetoma** | | |
| Actinomycetoma | 71 | 36.8 |
| Eumycetoma | 91 | 47.2 |
| Undetermined | 31 | 16.1 |

(1), *Fusarium solani* (1), and *Pseudallescheria boydii (anamorphic*: *Scedosporium boydii/Scedosporium boydii)* (1).

X-Ray examination was done in all the patients and was normal in the majority of cases 114 (59.1%). Bone lesions were identified in 79 patients (40.9%).

At interview, 156 (80.8%) patients have reported history of phytotherapy including 60 cases who have used oral plus local medicinal plants (Table 3), however the nature of the plants used was not mentioned. Mycetoma, clinical cases were treated regarding the type of mycetoma and the availability of the drug. Most of actinomycetoma cases were treated with trimethoprim-sulfamethoxazole, or a combination of trimethoprim-sulfamethoxazole, streptomycin, and amoxicillin/clavulanic acid (Table 3). In eumycetoma cases, itraconazole and terbinafine were used as antifungal agent combined in most cases with surgical treatment as shown in Table 3. Surgical treatment consisted of wide local excision and amputation (Table 3).

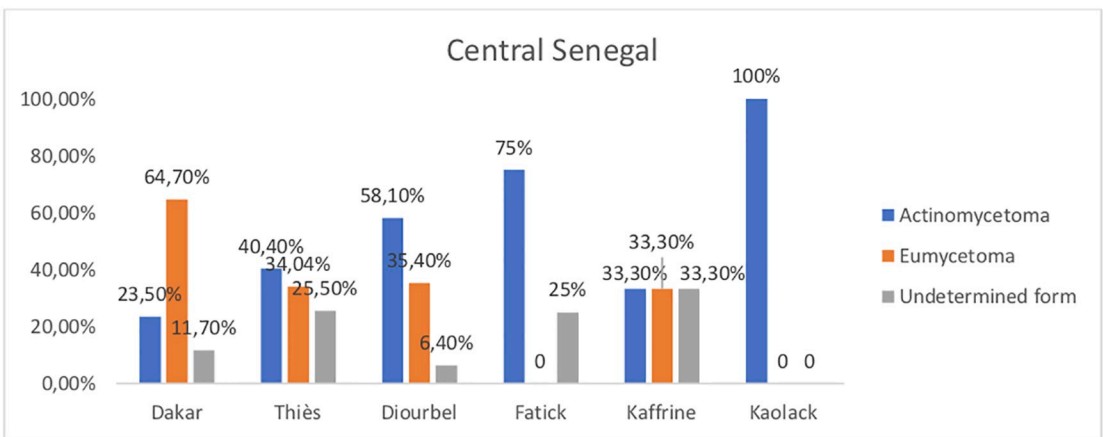

A

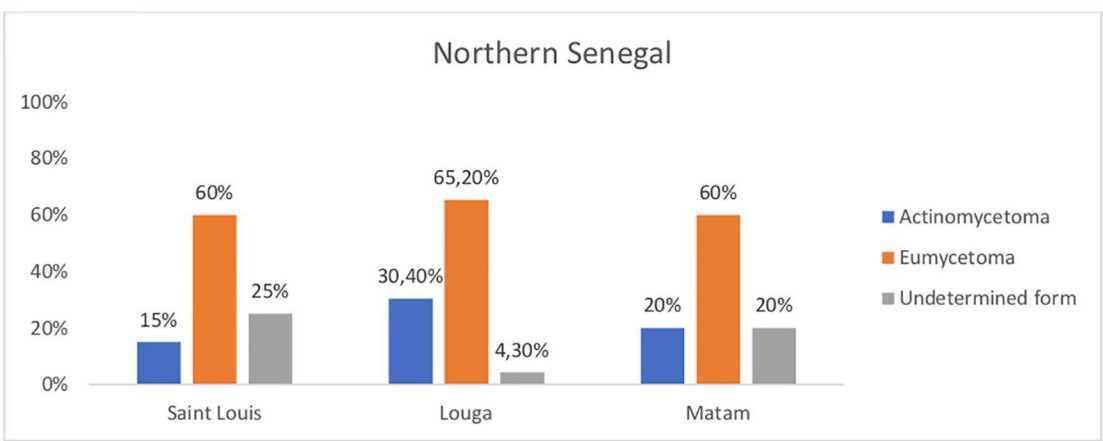

B

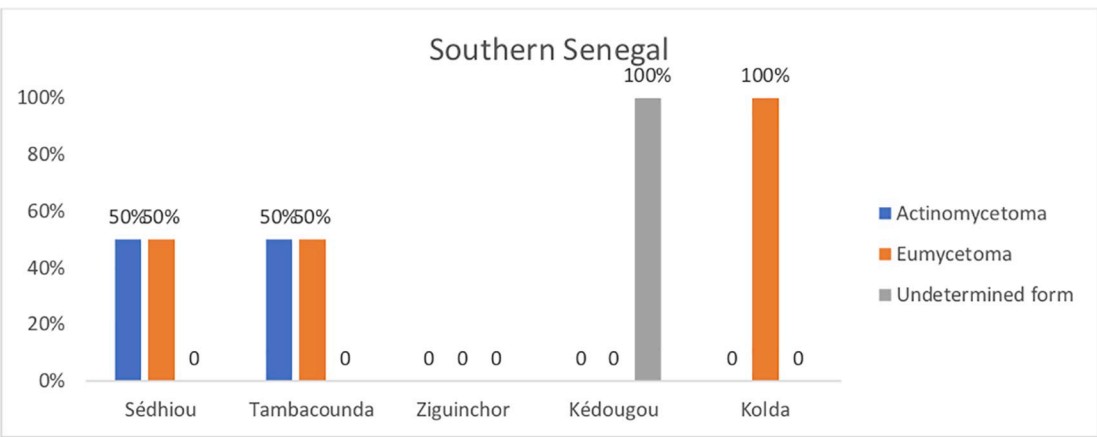

C

**Fig 1. Distribution of mycetoma types across Senegal.**

**Table 2. Demographic and clinical findings according to the type of mycetoma.**

| | Actinomycetoma N = 71 | EumycetomaN = 91 | Undetermined N = 31 | Total N (%) | p-value |
|---|---|---|---|---|---|
| Age in years, mean (SD) | 40.1 (16.3) | 35.5 (14.8) | 42.5 (19.5) | | 0.06 |
| Male /Female ratio | 2.9 | 4.05 | 1.3 | | |
| **Clinical aspects, n (%)** | | | | | |
| **Evolution** | | | | | |
| < 1 year | 4(5.7) | 2(2.2) | 1(3.2) | 7 (3.6) | 0.79 |
| 1–5 years | 33(46.5) | 40(44) | 17(54.9) | 90 (46.6) | |
| 5–10 years | 23(32.3) | 29(31.8) | 7(22.6) | 59 (30.6) | |
| 10–20 years | 7(9.8) | 16(17.6) | 5(16.1) | 28 (14.5) | |
| > 20 years | 4(5.7) | 4(4.4) | 1(3.2) | 9 (4.6) | |
| **Presence of pain** | | | | | |
| Yes | 41(57.7) | 44(48.3) | 17(54.9) | 102 (52.8) | 0.47 |
| No | 30(42.3) | 47(51.7) | 14(45.1) | 91 (47.1) | |
| **Clinical forms** | | | | | |
| Tumorous form | 28(39.4) | 40(43.9) | 14(45.1) | 82 (42.5) | 0.20 |
| Inflammatory form | 37(52.1) | 41(45.1) | 10(32.3) | 88 (45.5) | |
| Cystic form without grain and fistula | 6(8.5) | 10(11) | 7(22.6) | 23 (11.9) | |
| **Size of lesion** | | | | | |
| <5 cm | 0 | 1(1.1) | 0 | 1 (0.5) | 0.01 |
| 5–10 cm | 12(17) | 6(6.6) | 4(12.9) | 22 (11.4) | |
| >10 cm | 53(74.6) | 59(64.8) | 17(54.8) | 129 (66.8) | |
| Undetermined | 6(8,4) | 25(27.5) | 10(32.3) | 41 (21.2) | |
| **Type of grains** | | | | | |
| Red | 41 (57.7) | 0 | 0 | 41 (21.2) | $<10^{-5}$ |
| Black | 0 | 73 (80.2) | 1 (3.3) | 74 (38.3) | |
| White | 15 (21.1) | 10 (11) | 17 (54.9) | 42 (21.8) | |
| Yellow | 5 (7.1) | 0 | 2 (6.4) | 7 (3.6) | |
| Absence of grains | 10(14.1) | 8(8.8) | 11 (35.4) | 29 (15.1) | |
| **Anatomical localization** | | | | | |
| Lower limbs (feet and leg) | 65(91.5) | 80(88) | 31(100) | 176 (91.2) | 0.28 |
| Upper Limb (Hand/Arm) | 0 | 4(4.3) | 0 | 4 (2.1) | |
| Trunk | 5(7.1) | 6(6.6) | 0 | 11 (5.6) | |
| Head and Neck | 1(1.4) | 1(1.1) | 0 | 2 (1.1) | |
| **Lymphatic dissemination** | | | | | |
| Yes | 32 (45.07) | 34 (37.3) | 6 (19.3) | 72 (37.3) | 0.01 |
| No | 34 (47.8) | 37 (40.6) | 19 (61.2) | 90 (46.6) | |
| ND | 5 (7.04) | 20 (21.9) | 6 (19.3) | 31 (16.1) | |
| **Bone lesions** | | | | | |
| Yes | 27 (38.02) | 41 (45.05) | 11 (35.4) | 79 (40.9) | 0.53 |
| No | 44 (61.9) | 50 (54.9) | 20 (64.5) | 114 (59.1) | |

Out of the 33 red-grain actinomycetoma cases treated with trimethoprim-sulfamethoxazole alone, 20 recovered (60.6% cure rate), 4 was lost to follow-up and 9 were still on treatment. Outcome was worse if these red-grain actinomycetoma cases were treated with trimethoprim-sulfamethoxazole + streptomycin + amoxicillin/clavulanic acid as out of the 38 patients, 10 recovered (26.3% cure rate), 2 had a recurrence, 4 was lost to follow-up and 22 were still on treatment.

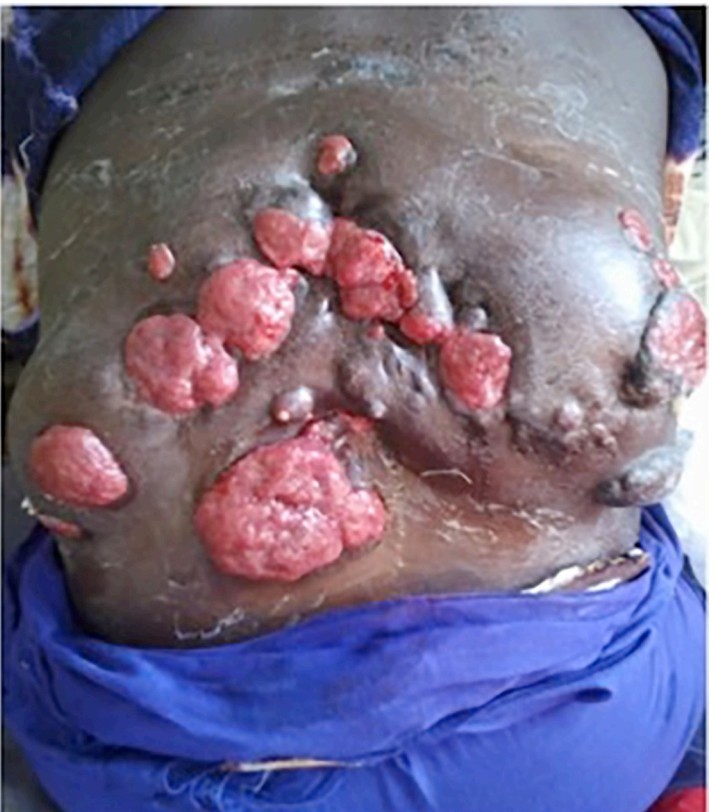

**Fig 2. A dorsolumbar tumoral actinomycetoma cases due to *Actinomadura pelletieri*.**

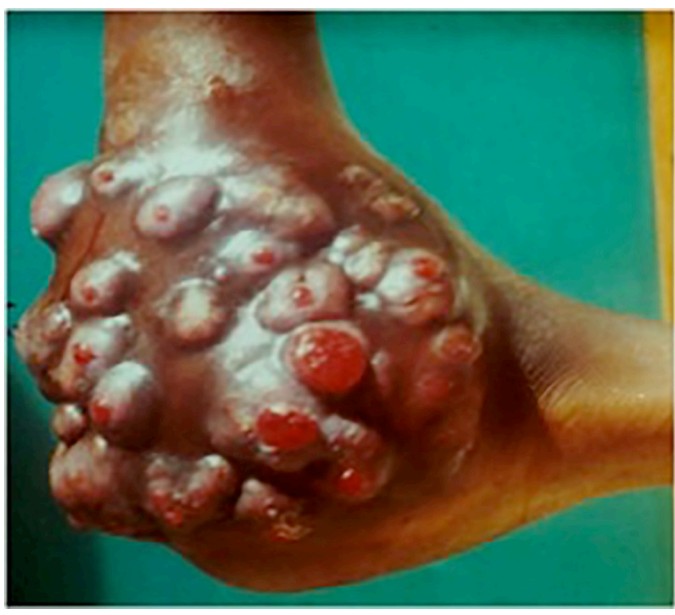

**Fig 3. "Madura foot" caused by *Actinomadura pelletieri*.**

**Table 3. Treatment and outcome according to the type of mycetoma.**

| | Actinomycetoma N = 71 | Eumycetoma N = 91 | Undetermined N = 31 | Total N (%) | p-value |
|---|---|---|---|---|---|
| **Traditional phytotherapy** | | | | | |
| Yes | 61(86) | 76(83.5) | 19(61.3) | 156 (80.8) | 0.009 |
| No | 10(14) | 15(16.5) | 12(38.7) | 37 (19.2) | |
| Local+Oral | 21(29.5) | 31(34.1) | 8(25.8) | 60 (31.1) | 0.91 |
| Local | 14(19.7) | 14(15.3) | 3(9.7) | 31 (16.1) | |
| Oral | 26(36.6) | 31(34.1 | 8(25.8) | 65 (28.5) | |
| **Medical treatment** | | | | | |
| Trimethoprim-sulfamethoxazole | 33(46.5) | 0 | 12(38.7) | 45(23.3) | $<10^{-5}$ |
| trimethoprim-sulfamethoxazole + Amox/Ac Clavulanique + Streptomycine | 38(53.5) | 0 | 4(12.9) | 42(21.8) | |
| trimethoprim-sulfamethoxazole + Amox/Ac Clavulanique + Itraconazole | 0 | 23(25.3) | 15(48.4) | 38(19.7) | |
| Terbinafine | 0 | 68(74.7) | 0 | 68(35.2) | |
| **Surgery** | | | | | |
| Yes | 9(12.7) | 78(85.7) | 13(41.9) | 100(51.8) | $<10^{-5}$ |
| No | 62(87.3) | 13(14.3) | 18(58.1) | 93(48.2) | |
| **Type of surgery** | | | | | |
| Resection | 5(7.1) | 43(47.2) | 6(19.3) | 54(27.9) | 0.02 |
| Amputation | 3(4.2) | 35(38.5) | 7(22.6) | 45(23.3) | |
| Lymph node dissection | 1(1.4) | 0 | 0 | 1(0.5) | |
| **Outcome** | | | | | |
| Full recovery | 30 (42.3) | 43 (47.3) | 6 (19.4) | 79(40.9) | $<10^{-5}$ |
| Recurrence | 2 (2.8) | 2 (2.2) | 3 (9.6) | 7(3.7) | |
| Lost to follow-up | 8 (11.3) | 14 (15.4) | 12 (38.7) | 34(17.6) | |
| Ongoing treatment | 31 (43.6) | 32 (35.1) | 10 (32.3) | 73(37.8) | |

Out of the white or yellow grain mycetoma cases, treatment with trimethoprim-sulfamethoxazole alone (12 patients) yielded 4 recovery (33.3% cure rate), 1 recurrence, 3 patients lost to follow-up and 4 cases still on treatment. Out of the 4 patients treated with the combination trimethoprim-sulfamethoxazole + streptomycin + amoxicillin/clavulanic acid, 2 was lost to follow-up and 2 were still on treatment. The 15 patients treated with Itraconazole+ antibiotics have presented 2 recovery (13% cure rate), 2 recurrence, 7 lost to follow-up and 4 cases still on treatment.

Out of the 68 black-grain eumycetoma cases treated with terbinafine and surgery, 20 recovered (29.4% cure rate), 2 had a recurrence, 14 were lost to follow-up and 32 were still on treatment. Outcome was better if these black-grain eumycetoma cases were treated with Itraconazole, trimethoprim-sulfamethoxazole + streptomycin + amoxicillin/clavulanic acid and surgery as all the 23 patients recovered (100% cure rate). The patients with eumycetoma cases were treated with terbinafine 500 mg twice daily for 24–48 weeks. In the 20 patients treated with terbinafine who recovered, one patient had a small lesion (less than 5 cm), 4 patients had lesions between 5–10 cm, and 10 patients had lesions larger than 10 cm.

Terbinafine treatment was combined with surgery in 49 of the 68 patients. Among these patients, 31 were treated with surgical removal while 18 were amputated. The 20 patients who recovered had been treated with terbinafine combined with surgery. Ten of them were treated with surgical removal and 10 were amputated. No patient has been cured with terbinafine only.

At the time of survey, 79 (40.9%) patients has presented a full recovery. Only 7(3.7%) patients has presented recurrence (Table 3).

## Discussion

With the view of strengthening mycetoma control, it is important to provide accurate and updated epidemiological information in endemic regions. This study provides, for the first time in Senegal, mycetoma cases series diagnosed in three hospital centres located in two different regions and allows an overview of the distribution of clinical cases. The mycetoma prevalence within the study period is underestimated because we have selected only well documented patients to strengthen the disease description.

The patients with mycetoma described in this series were young adults in general with an mean age of 38 years, which is in accordance with previous reports in Senegal and many other countries that have shown the higher frequency of mycetoma between 20 and 40 years of age [11,12,15,16]. However, 35.2% of our patients were more than 45 years old. This category of patients was also the most infected group in a series reported in Brazil with an mean age of 48 years [17]. The excess of male patients found in this study is in agree with the literature [1,3,5,11]. Some authors have suggested the role of hormonal and genetics factors in this elevated male: female ratio [18,19].

Patients were more frequently diagnosed with eumycetoma (47.2%) than actinomycetoma (37.8%) in our study. In contrast, previous studies in Senegal have reported a relatively higher prevalence of actinomycetoma in their series [11,16]. This discrepancy might be explained by a recruitment bias because most of these data originated from the Le Dantec hospital dermatology ward where actinomycetoma form are usually managed. However, the predominance of eumycetoma cases in our series is in agree with the findings of Ndiaye *et al.*, who have reported 70% eumycetoma *versus* 30% actinomycetoma in Senegal [12]. Other reports confirm the relatively high prevalence of eumycetoma in the West-African region [20,21]. Most of cases in our study were diagnosed based on clinical aspects and grain color. Only half of the cases were confirmed by mycological techniques and histopathology. Indeed, the diagnosis and the confirmation of mycetoma cases is challenging in our resources limited settings. In Senegal, culture and histopathology are the gold standard methods in hospital settings. However, both techniques are operator dependent and need experience which could have its reflections on the accuracy. Many difficulties have been noted in our laboratory settings, including negative culture, the misidentifications of the causative fungal agents and the inability of the technicians to correctly describe the histopathological appearance of mycetoma causative agents. This situation emphasized the need for capacity building of our laboratory technicians on mycetoma diagnosis.

Despite the use of diverse laboratory techniques coupled with clinical findings, the etiological agent remained undetermined in 16% of the patients in our study. This is quite similar to Fahal *et al.*, where the etiological agent was undetermined in 13% of their patients [3]. The difficulty of identifying the etiological agents can be explained by the lack of adequate diagnostic tools in most of our endemic countries and the impossibility in certain cases to differentiate the causative agents using the available mycological and histological techniques [22,23]. For example, it is well known that histological features cannot differentiate *Acremonium spp* from *Fusarium spp*, which both produce yellow to white grains with heterogeneous histological aspects [2]. The direct nucleotide sequencing in biopsy samples of the 16S rRNA gene, to document actinomycetoma, and the rRNA gene internal transcribed spacer regions, to document eumycetoma, have been proposed to enhance pathogen identification capacities [24–28].

The geographical distribution of the patients in our study revealed the predominance of eumycetoma in the Northern region of Senegal and of actinomycetoma in the Central and Southern part of the country is in agreement with previous reports in the country [10–12]. This spatial distribution is probably associated with eco-climatic factors, because the dry tropical climate in the North contrasts to the higher annual rainfall in the South. The effect of environmental factors and climate on the distribution of mycetoma cases have been demonstrated in many other countries within the African continent and elsewhere [3,15,29–31]. It is important to note the relatively high number of patients originating from the capital-city Dakar in our study, which is considered as a non-endemic area. One explanation might be the usually protracted incubation lag of the disease; the patients had probably been infected before immigrating to the capital-city. Another likely explanation is that specialized health-care facilities that are capable to manage mycetoma are mainly available in the capital-city.

Patients with mycetoma usually attend the clinic at late stage of the disease [32]. Accordingly, most of the times the lag between the disease onset and hospital attendance exceeded 10 years in many of the patients in our study. Most of the patients seen in our study presented with local pain, which is probably associated with an advanced stage of the disease. This late presentation at the hospital can be explained by the lack of capacity among health workers, particularly in rural area, and consequently the absence of health education among patients and [33,34]. Most of the patients in this study had used of traditional medicine before attending specialized clinics. This is in line with previous reports [35].

The clinical presentation was quite similar in both actinomycetoma and eumycetoma; however, bone lesions appeared at an earlier stage in actinomycetoma than in eumycetoma. In this study, most of the lesions was tumor-like, irrespective of the mycetoma type. The lower limbs (feet and leg) were the most affected sites as described in many previous studies [1,3,11,12]. However, it is important to note that 37.3% of lesions have spread to the lymph nodes and 40.9% have reached the bones. The frequency of lymphadenopathy is relatively high in this series compared to reports from Sudan (10.3%) or Mexico (1.65%) [1,3]. It is well known that mycetoma may spread to the lymph nodes particularly in case of multiple inadequate surgery [33,36,37]. Some authors have also demonstrated the possibility of blood borne spread in certain cases. This situation emphasized the need to train surgeons on the management of mycetoma and to highlight that local anaesthesia is contraindicated in mycetoma [7].

Most of the mycetoma treatment used in this study is in line with previous published reports. For actinomycetoma, trimethoprim-sulfamethoxazole administrated alone gave better results (60.6% cure rate) compared to the combination of trimethoprim-sulfamethoxazole and the other antibiotics. However, it was difficult to give any conclusion in this last group as most of the patients was still on treatment (trimethoprim-sulfamethoxazole + streptomycin + amoxicillin/clavulanic acid) at the time of this study. Indeed, several studies have shown the excellent clinical response of the combination Co-trimoxazole and Amikacin Sulfate achieving in some studies a cure rate of about 90% [38,39]. So, this combination remains the recommended treatment for actinomycetoma cases.

For black grain eumycetoma cases, terbinafine was used in the majority of the patients in this series with a low cure rate (29.3%) compared to the excellent response obtained with the combination itraconazole and antibiotics. These results are in line with previous reports showing the low response of eumycetoma to terbinafine. For example, Ndiaye et al have reported 25% cure rate in 23 patients treated with terbinafine [40]. However, our study has some limitations as several patients did not completed their follow-up. So, there is a need for further studies to compare the efficacy of these two drugs.

Another big challenge in the control of mycetoma is the management cost particularly in eumycetoma. Treatments are expensive and several patients do not continue their follow up.

Therefore, there is a need to find innovative and cost-effective strategies to manage mycetoma cases in our endemic settings.

In conclusion, this study has highlighted the limitations of mycetoma diagnosis in Senegal, the delayed access to healthcare and treatment, and the lack of adequate therapeutic resources. The limitations on fungal species identification is also a challenge as the management and outcome of cases depends on the causative agent. The baseline data provided will contribute to the establishment of a surveillance system for better control and advocate for more resources to fight this devastating disease.

## Author Contributions

**Conceptualization:** Doudou Sow, Maodo Ndiaye, Babacar Faye.

**Data curation:** Doudou Sow, Babacar T. Faye.

**Formal analysis:** Stéphane Ranque.

**Investigation:** Doudou Sow, Maodo Ndiaye, Lamine Sarr, Mamadou D. Kanté, Fatoumata Ly, Pauline Dioussé, Abdou Magip Gaye, Cheikh Sokhna.

**Software:** Babacar T. Faye.

**Supervision:** Doudou Sow, Mamadou D. Kanté, Babacar Faye.

**Validation:** Babacar T. Faye, Stéphane Ranque.

**Writing – original draft:** Doudou Sow, Stéphane Ranque.

**Writing – review & editing:** Doudou Sow, Maodo Ndiaye, Lamine Sarr, Mamadou D. Kanté, Fatoumata Ly, Pauline Dioussé, Babacar T. Faye, Abdou Magip Gaye, Cheikh Sokhna, Stéphane Ranque, Babacar Faye.

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
