## [Editor Report · Decision Letter 0]

18 Mar 2020

PONE-D-20-06483

Mycetoma epidemiology, diagnosis management, and outcome in three hospital centres in Senegal from 2008 to 2018

PLOS ONE

Dear Dr. SOW,

Thank you for submitting your manuscript to PLOS ONE. After careful consideration, we feel that it has merit but does not fully meet PLOS ONE’s publication criteria as it currently stands. Therefore, we invite you to submit a revised version of the manuscript that addresses the points raised during the review process.

We would appreciate receiving your revised manuscript by April 1 2020 11:59 PM. To enhance the reproducibility of your results, we recommend that if applicable you deposit your laboratory protocols in protocols.io, where a protocol can be assigned its own identifier (DOI) such that it can be cited independently in the future. For instructions see: http://journals.plos.org/plosone/s/submission-guidelines#loc-laboratory-protocols

We look forward to receiving your revised manuscript.

Kind regards,

Abdallah M. Samy, PhD

Academic Editor

PLOS ONE

**Additional Editor Comments:**

The authors didn't address the previous comments by Reviewer #2. Please check the decision letter for your revised manuscript sent by PLOS NTDs and address all comments raised by reviewer #2 (i.e. I added a copy of these comments in this email below). This step is necessary before considering your manuscript for further processing in PLOS ONE. Thanks too much! AMS

Journal Requirements:

2. We note that Figures 1-2 in your submission contain [map/satellite] images which may be copyrighted. All PLOS content is published under the Creative Commons Attribution License (CC BY 4.0), which means that the manuscript, images, and Supporting Information files will be freely available online, and any third party is permitted to access, download, copy, distribute, and use these materials in any way, even commercially, with proper attribution. For these reasons, we cannot publish previously copyrighted maps or satellite images created using proprietary data, such as Google software (Google Maps, Street View, and Earth). For more information, see our copyright guidelines: http://journals.plos.org/plosone/s/licenses-and-copyright.

1.    You may seek permission from the original copyright holder of Figures 1-2 to publish the content specifically under the CC BY 4.0 license. 

**Reviewers' comments:**

**Methods**

-Are the objectives of the study clearly articulated with a clear testable hypothesis stated?

-Is the study design appropriate to address the stated objectives?

-Is the population clearly described and appropriate for the hypothesis being tested?

-Is the sample size sufficient to ensure adequate power to address the hypothesis being tested?

-Were correct statistical analysis used to support conclusions?

-Are there concerns about ethical or regulatory requirements being met?

Reviewer #2: The revised manuscript was not there. Only in the marked-up version I saw changes made. This made it difficult to review. I assume that the marked-up version is the revised version.

Line 126: add the following: at the time of this study, molecular diagnosis was not available at any of the centers. Therefore species identification could not be confirmed molecularly.

Line 127: Add: medical treatment consisted of ... or ... for actinomycetoma and ... or ... for eumycetoma. In case it was not known if the lesion was an actino- or an eumycetoma, patients were treated with ... Eumycetoma were also treated surgically.

**Results**

-Does the analysis presented match the analysis plan?

-Are the results clearly and completely presented?

-Are the figures (Tables, Images) of sufficient quality for clarity?

Reviewer #2: Results:the paragraph on the treatment outcome is great but not complete. Could you also tell something on the other treatments and calculate a cure rate. Readers would like to know especially for terbinafine if this is comparable to itraconazole. I know you treated itracanazole in combination with antibiotics, however there is a study from Sudan where this was compared to the classical treatment with itraconazole alone. You could use this in your discussion to speculate what in your patient population was the best terbinafine or itraconazole.

E.g. Out of the x red-grain actinomycetoma cases treated with Bactrim only, … recovered, … had a recurrence and .. are still on treatment. Outcome was similar/better/worse if these red-grain actinomycetoma cases were treated with Bactrim + … as … recovered, … had a recurrence and .. are still on treatment. Out of the white or yellow grain actinomycetoma cases ….

Out of the … black grain eumycetoma cases treated with terbinafin and surgery, … recovered, … had a recurrence and .. are still on treatment. Outcome was similar/better/worse if these black-grain actinomycetoma cases were treated with Itraconazole, Bactrim and Clavulanique+ Streptomycine and suergy. For those patients … recovered, … had a recurrence and .. are still on treatment.

**Conclusions**

-Are the conclusions supported by the data presented?

-Are the limitations of analysis clearly described?

-Do the authors discuss how these data can be helpful to advance our understanding of the topic under study?

-Is public health relevance addressed?

Reviewer #2: Yes. However in the discussion paragraph 284-288 a little bit more discussion is needed on the treatment.

**Editorial and Data Presentation Modifications?**

Reviewer #2: Line 91: actinomycete

Line 95: the surgical treatments range from broad surgical ...

Line 97: this recognition has brought ..

Line 166: In 102 cases (52.8%) the patients experienced pain. In ... of these patients a painkiller was added to their treatment.

Line 183: add to the sentence: At interview, 156 (80.8%) patients have reported history of phytotherapy including 60 cases 186 who have used oral plus local medicinal plants, however the nature of the plants used was not mentioned. Mycetoma, clinical cases were treated …

Line 189: what is the 100 referring to? I did not see it in the table. Please remove this and say … most cases with surgical treatment as shown in table 3.

Line 248. You mention that most of the patients presented local pain. Did you give them something to ease the pain? E.g. diclofenac? In the study of Dupont et al. they used diclofenac to ease the pain in an advanced eumycetoma case. They added it to the posaconazole treatment. Unexpectedly, the lesion started healing and the mycetoma disappeared. Did you ever experience this in your patients?

**Summary and General Comments**

Reviewer #2: I think it is an important study and needs to be published. However the language still needs polishing up and some adjustments are still needed, especially on the treatment outocome.

---

## [Author Response · Author response to Decision Letter 0]

24 Mar 2020

METHODS

Reviewer #2: The revised manuscript was not there. Only in the marked-up version I saw changes made. This made it difficult to review. I assume that the marked-up version is the revised version.

Line 126: add the following: at the time of this study, molecular diagnosis was not available at any of the centers. Therefore, species identification could not be confirmed molecularly.

Line 127: Add: medical treatment consisted of ... or ... for actinomycetoma and ... or ... for eumycetoma. In case it was not known if the lesion was an actino- or an eumycetoma, patients were treated with ... Eumycetoma were also treated surgically.

1. Response: These clarifications have been added in the manuscript as recommended. Line 139 to 148 in the clean revised version

RESULTS

Reviewer #2: Results: the paragraph on the treatment outcome is great but not complete. Could you also tell something on the other treatments and calculate a cure rate. Readers would like to know especially for terbinafine if this is comparable to itraconazole. I know you treated itracanazole in combination with antibiotics, however there is a study from Sudan where this was compared to the classical treatment with itraconazole alone. You could use this in your discussion to speculate what in your patient population was the best terbinafine or itraconazole.

E.g. Out of the x red-grain actinomycetoma cases treated with Bactrim only, … recovered, … had a recurrence and .. are still on treatment. Outcome was similar/better/worse if these red-grain actinomycetoma cases were treated with Bactrim + … as … recovered, … had a recurrence and .. are still on treatment. Out of the white or yellow grain actinomycetoma cases ….

Out of the … black grain eumycetoma cases treated with terbinafin and surgery, … recovered, … had a recurrence and .. are still on treatment. Outcome was similar/better/worse if these black-grain actinomycetoma cases were treated with Itraconazole, Bactrim and Clavulanique+ Streptomycine and suergy. For those patients … recovered, … had a recurrence and .. are still on treatment.

2. Response: This paragraph including the details of treatment and cure rates has been added in the result section as recommended. Line 214 to 231 in the clean revised version

About the discussion to speculate on the best treatment, see response 3.

CONCLUSIONS

Reviewer #2: Yes. However, in the discussion paragraph 284-288 a little bit more discussion is needed on the treatment.

3. Response: A paragraph discussing the treatment has been added in the discussion paragraph as recommended, line 324 to 338 in the clean revised version. Three more references have been added in the reference list. 

EDITORIAL AND DATA PRESENTATION MODIFICATIONS?

Reviewer #2: Line 91: actinomycete

Line 95: the surgical treatments range from broad surgical ...

Line 97: this recognition has brought ..

Line 166: In 102 cases (52.8%) the patients experienced pain. In ... of these patients a painkiller was added to their treatment.

Line 183: add to the sentence: At interview, 156 (80.8%) patients have reported history of phytotherapy including 60 cases 186 who have used oral plus local medicinal plants, however the nature of the plants used was not mentioned. Mycetoma, clinical cases were treated …

Line 189: what is the 100 referring to? I did not see it in the table. Please remove this and say … most cases with surgical treatment as shown in table 3.

Line 248. You mention that most of the patients presented local pain. Did you give them something to ease the pain? E.g. diclofenac? In the study of Dupont et al. they used diclofenac to ease the pain in an advanced eumycetoma case. They added it to the posaconazole treatment. Unexpectedly, the lesion started healing and the mycetoma disappeared. Did you ever experience this in your patients?

4. Response: The above corrections have been made in the manuscript as recommended. 

Line 91 (Old version) <-> Line 104 (revised version)

Line 95: (Old version) <-> Line 107 (revised version)

Line 97: (Old version) <-> Line 110 (revised version)

Line 166: (Old version) <-> Line 187 (revised version)

Line 183: (Old version) <-> Line 205 (revised version)

Line 189: (Old version) <-> Line 211 (revised version)

Line 248: About the pain, the patients did not receive diclofenac, they have just used paracetamol.

SUMMARY AND GENERAL COMMENTS

Reviewer #2: I think it is an important study and needs to be published. However, the language still needs polishing up and some adjustments are still needed, especially on the treatment outcome.

5. Response: The adjustments on the treatment requested by the reviewer have been done in the results (See response 2). Corrections

JOURNAL REQUIREMENTS:

2. We note that Figures 1-2 in your submission contain [map/satellite] images which may be copyrighted. All PLOS content is published under the Creative Commons Attribution License (CC BY 4.0), which means that the manuscript, images, and Supporting Information files will be freely available online, and any third party is permitted to access, download, copy, distribute, and use these materials in any way, even commercially, with proper attribution. For these reasons, we cannot publish previously copyrighted maps or satellite images created using proprietary data, such as Google software (Google Maps, Street View, and Earth). For more information, see our copyright guidelines: http://journals.plos.org/plosone/s/licenses-and-copyright.

6. Response: Figures 1-2 have been removed. Data in Fig 1 are presented in table 1. The map (Fig2) showing the distribution of mycetoma types has been replaced by a histogram of the cases in the northern, central and southern of Senegal (New figure 1). Figures 3 and 4 have been renamed Fig 2 and 3.

---

## [Editor Report · Decision Letter 1]

3 Apr 2020

Mycetoma epidemiology, diagnosis management, and outcome in three hospital centres in Senegal from 2008 to 2018

PONE-D-20-06483R1

Dear Dr. SOW,

We are pleased to inform you that your manuscript,  "Mycetoma epidemiology, diagnosis management, and outcome in three hospital centres in Senegal from 2008 to 2018" (PONE-D-20-06483R1), has been judged scientifically suitable for publication and will be formally accepted for publication once it complies with all outstanding technical requirements.

With kind regards,

Abdallah M. Samy, PhD

Academic Editor

PLOS ONE

Additional Editor Comments:

Please consider the final version attached with this decision letter for production; there are some corrections in the tables. Thanks! AMS

---

## [Editor Report · Acceptance letter]

10 Apr 2020

PONE-D-20-06483R1 

Mycetoma epidemiology, diagnosis management, and outcome in three hospital centres in Senegal from 2008 to 2018 

Dear Dr. Sow:

I am pleased to inform you that your manuscript has been deemed suitable for publication in PLOS ONE. Congratulations! Your manuscript is now with our production department. 

With kind regards,

on behalf of

Dr. Abdallah M. Samy 

Academic Editor

PLOS ONE